# Antibacterial Activity of Dental Composite with Ciprofloxacin Loaded Silver Nanoparticles

**DOI:** 10.3390/molecules27217182

**Published:** 2022-10-24

**Authors:** Wafa Arif, Nosheen Fatima Rana, Iqra Saleem, Tahreem Tanweer, Muhammad Jawad Khan, Sohad Abdulkaleg Alshareef, Huda M. Sheikh, Fatima S. Alaryani, Manal Othman AL-Kattan, Hanan Ali Alatawi, Farid Menaa, Aroosa Younis Nadeem

**Affiliations:** 1Department of Biomedical Engineering and Sciences, School of Mechanical & Manufacturing Engineering, National University of Sciences & Technology, Islamabad 44000, Pakistan; warif.bmes19smme@student.nust.edu.pk (W.A.); isaleem.bmes19smme@student.nust.edu.pk (I.S.); ttanweer.phd19smme@student.nust.edu.pk (T.T.); ayounis.phd21smme@student.nust.edu.pk (A.Y.N.); 2Integrated Nanobiotechnology Laboratory, School of Interdisciplinary Engineering & Sciences (SINES), National University of Sciences and Technology (NUST), Islamabad 44000, Pakistan; 3Department of Robotics & Artificial Intelligence, School of Mechanical & Manufacturing Engineering, National University of Science & Technology, Islamabad 44000, Pakistan; jawad.khan@smme.nust.edu.pk; 4Department of Chemistry, Faculty of Science, University of Tabuk, Tabuk 71491, Saudi Arabia; s_alshareef@ut.edu.sa; 5Department of Biology, College of Science, University of Jeddah, Jeddah 21589, Saudi Arabia; hmsheikh@uj.edu.sa (H.M.S.); fsalaryani@uj.edu.sa (F.S.A.); moalkattan8@uj.edu.sa (M.O.A.-K.); 6Department of Biological Sciences, University College of Haqel, University of Tabuk, Tabuk 71491, Saudi Arabia; halatwi@ut.edu.sa; 7Internal Medicine and Nanomedicine, California Innovations Corporation, San Diego, CA 92037, USA

**Keywords:** ciprofloxacin, dental composite, nanoparticles, antibacterial activity

## Abstract

Resin composites have been widely used in dental restoration. However, polymerization shrinkage and resultant bacterial microleakage are major limitations that may lead to secondary caries. To overcome this, a new type of antibacterial resin composite containing ciprofloxacin-loaded silver nanoparticles (CIP-AgNPs) were synthesized. The chemical reduction approach successfully produced CIP-AgNPs, as demonstrated by FTIR, zeta potential, scanning electron microscopy, and ultraviolet-visible (UV-vis) spectroscopy. CIP-AgNPs were added to resin composites and the antibacterial activity of the dental composite discs were realized against *Enterococcus faecalis*, *Streptococcus mutans*, and the Saliva microcosm. The biocompatibility of modified resin composites was assessed and mechanical testing of modified dental composites was also performed. The results indicated that the antibacterial activity and compressive strength of resin composites containing CIP-AgNPs were enhanced compared to the control group. They were also biocompatible when compared to resin composites containing AgNPs. In short, these results established strong ground application for CIP-AgNP-modified dental composite resins.

## 1. Introduction

Resin composites are commonly used in dental restoration procedures and are preferred due to their aesthetic appearance and acceptable physiochemical properties [1,2,3]. However, there are certain disadvantages of this class of restorative material [1,2,3]. One major limitation is polymerization shrinkage, which creates a gap between the tooth and the restorative material leading to microleakage and biofilm formation that leads to secondary caries [1,2,3]. Secondary caries, if not treated, eventually lead to dental restoration failure [4,5]. To counter this, novel therapeutical composites exhibiting antibacterial properties were developed [6]. For the first time in the 1950s, antibiotic drugs were incorporated into dental resin composites to enhance their antibacterial potential [7]. Since then, the focus has been shifted towards this approach and multiple antibacterial agents belonging to different classes have been utilized, for example chlorhexidine (CHX), chitosan, silver, etc. [8,9].

The application of nanoparticles (NPs) with antimicrobial properties has great potential in this respect [10]. In dentistry, metallic NPs have been widely used due to their high antibacterial activity [11]. They have been added within resin composites to enhance their antibacterial properties and to reduce the reoccurrence of caries [11]. Among metallic NPs, silver nanoparticles (AgNPs) are of great importance because of their high antibacterial potential [12]. They target the bacterial membrane and disrupt metabolic processes in bacterial cells, leading to cell death [13]. These NPs have been widely utilized as drug delivery systems, where in combination with antibiotics they impart potent antibacterial activity [7,8].

Ciprofloxacin (CIP) is a broad-spectrum fluoroquinolone that is widely utilized for the treatment of multiple bacterial infections [14]. CIP has been loaded onto multiple nanoplatforms, including CIP-loaded nanocomposites, CIP-loaded polymeric nanofibers, CIP-loaded nanotubes, and CIP-loaded AgNPs [15,16,17]. It is reported that when loaded onto the AgNPs, CIP showed enhanced antibacterial activity against Gram-positive and Gram-negative bacteria [18].

CIP-AgNPs have not been previously reported for their application as antibacterial activity enhancers in dental composites. In this work, dental resin composites integrated with CIP-AgNPs were synthesized and their composite discs with these NPs were then investigated against bacterial strains of *Enterococcus faecalis*, *Streptococcus mutans*, and Saliva microcosm. The effect of CIP-AgNPs on the mechanical strength and biocompatibility of the resin composite was also realized.

## 2. Results

### 2.1. Synthesis of AgNPs & Ciprofloxacin Loaded AgNPs

UV analysis was performed for AgNPs and CIP-AgNPs. Within the 400–420 nm region, AgNPs showed a very strong and wide peak at 404 nm. The peak shifted to 414 nm on CIP loading. This peak shift demonstrated effective loading of CIP on AgNPs (Figure 1).

### 2.2. Structural Analysis of CIP-AgNPs by FTIR Spectroscopy

There were characteristic peak shifts in CIP-AgNPs. The stretching of hydroxide and hydrogen bonding caused a peak at 3340 cm^−1^ in the FT-IR spectra of the CIP-AgNPs. The C-F stretching vibration caused a peak at 1085 cm^−1^ (Table 1, Figure 2).

### 2.3. Surface Morphology of AgNPs and Ciprofloxacin-Loaded AgNPs

The shape and size of prepared NPs were analyzed by SEM. at 20 kV (Figure 3). SEM images revealed that the NPs were spherical. An increase in the average size of the AgNPs from 33.5 nm to 98.32 nm showed successful loading of CIP.

### 2.4. Zeta Potential of Nanoparticles

Loading the AgNPs with CIP changed the zeta potential from −14.5 mV to +15.8 mV. Hence, it imparted a positive charge on the NP’s surface (Figure 4).

### 2.5. In Vitro Antibacterial Activity of Dental Composites

CIP-AgNPs exhibited a higher antibacterial activity than pure resin composite and pure AgNPs for all three models against *E. faecalis*, *S. mutans*, and Saliva microcosm. The resin composite discs containing 1% CIP-AgNPs exhibited significantly higher antibacterial activity than the control groups that were pure resin composite discs and composite discs containing 1% AgNPs. Since composite discs containing 1% CIP-AgNPs presented almost clear plates, it is evident that even a content 1% CIP-AgNPs provides a very strong antibacterial agent.

#### Antibacterial Activity

The antibacterial activity of composite discs loaded with AgNPs and CIP-AgNPs is presented in Figure 5, Figure 6 and Figure 7. Compared to the bacterial counts of AgNP-modified and unmodified composite discs, there was a striking decrease in the bacterial counts of CIP-AgNP-modified composite resin discs. Saliva microcosm, *Streptococcus mutans*, and *Enterococcus faecalis* showed significantly less bacterial growth when exposed to the composite discs containing 1% CIP-AgNPs (*p* < 0.05) (Figure 8).

### 2.6. Biocompatibility Analysis 

The Figure 9 presents the hemolytic activity of the pure dental resin composite discs and disks with AgNPs, and CIP-AgNPs. According to ISO/TR 7406 if any material’s percentage hemolysis value is less than 5, it is considered safe. AgNPs showed hemolytic activity, which was more toxic than CIP-AgNPs. The ideal concentration for AgNPs is 1%. On the other hand, cytotoxicity was dramatically decreased (*p* < 0.05) when CIP was loaded into AgNPs. With the addition of 1% CIP-AgNPs, there was little to no noticeable hemolytic activity. These results show that composite discs containing CIP-AgNPs were more more biocompatible than those containing 1% AgNPs. 

### 2.7. Compressive Strength

When CIP-AgNPs were added to resin composites at 1% concentration, the compressive strength of the composite resin discs with CIP-AgNPs significantly increased (*p* < 0.05) compared to the composite resin discs containing 1% AgNPs (Figure 10). The maximum compressive strength was shown by a composite containing 1% CIP-AgNPS. This showed that adding CIP-AgNPs did not negatively affect dental composite discs’ mechanical properties. It enhanced the compressive strength of dental composite discs.

## 3. Discussion

CIP is a broad-spectrum antibiotic that is highly effective for *Streptococcus* species and *Enterococcus* species [REF] CIP was loaded on AgNPs to enhance the antibacterial activity and mechanical properties.

The current study showed that with the addition of CIP on AgNPs, the absorption maxima have shifted from 404 nm to 414 nm, which is close to the initial peak. This phenomenon declared that AgNPs were successfully capped with Ciprofloxacin.

AgNPs exhibited the zeta potential of −14.5 mV, while CIP-AgNPs exhibited the zeta potential of +15.8 mV. The opposite charges supported the electrostatic interaction between silver nanoparticles and CIP-AgNPs, which led to the successful loading of Ciprofloxacin on silver nanoparticles. 

SEM showed circular-shaped CIP-AgNPs, which are almost three times bigger due to the coating of Ciprofloxacin. CIP-AgNPs were aggregated due to low positive charge. FTIR analysis was performed to understand the stabilizing of CIP on AgNPs, demonstrating the spectra of CIP-AgNPs, C.I.P., and AgNPs recorded at the region of 400–4000 cm^−1^. The peak at 1380 cm^−1^ shifted to 1405 cm^−1^ in CIP-AgNPs due to -NH group protonation in CIP, confirming the electrostatic interaction between CIP and AgNPs. The peak at 3530 cm^−1^ shifted to 3340 cm^−1^ due to hydrogen bonding between the OH group of CIP and the borohydride group of AgNPs There were significant shifts observed in the FTIR spectra of CIP and CIP-AgNPs [19]. The characteristic peak at 1085 cm^−1^ was due to the C-F stretching of CIP [20].

The composite discs with CIP-AgNPs showed improved antibacterial activity. The results of this study are consistent with [21], which claims that adding small amounts of AgNPs to composite resin produced a stronger antibacterial effect than unmodified resin. Lower concentrations of AgNPs showed less toxicity and greater mechanical strength. Composite resins containing 1% AgNPs reduced *S. mutans* and *E. faecalis* growth. The toxicity and mechanical strength of composite resin would be impacted by an increase in the concentration of AgNPs above 1% [22]. This research demonstrated that 1% CIP-AgNPs showed a much-enhanced antibacterial effect compared to 1% AgNPs without very little cellular toxicity and exhibited more mechanical strength of composite resin. They inhibited *S. mutans* and *E. faecalis* growth after being added to composite resin (*p* < 0.05). AgNPs had shown around 6% hemolytic activity. The concentration of 1% AgNPs is indeed optimal [23]. When CIP was added to AgNPs, the cytotoxicity was significantly decreased. 

### 3.1. Mechanism of Action

The metal Ag has a wide range of activities against bacteria, fungus, and some viruses, and is much improved when it is present in nano form. AgNPs have an attraction for sulfur- and phosphorus-containing groups that are present in the bacterial membranes. When incorporated into the resin composites, these NPs may cause transmembrane electron transport to be damaged by the release of Ag ions, which prevents DNA replication [24]. DNA gyrase enzyme and topoisomerase IV activity have been shown to be hindered after loading the AgNPs with CIP. This allows antibiotic entrance into the bacterial cells, ultimately causing bacterial cell death [25].

A good dental composite should have superior mechanical features and better antibacterial properties. The current study shows that the loading of CIP on AgNPs improved the antibacterial activity and enhanced the compressive strength significantly. This investigation showed that adding CIP-AgNPs improved the mechanical characteristics and antibacterial activity of resin-based composite restorations.

### 3.2. Limitation 

There are some limitations of the study that needs to be addressed. The loading capacity of CIP on AgNPs and particle size distribution (PSD) in composite have not been reported; hence these parameters should be carried out in future studies. Also, composites with only CIP as positive control have not been evaluated for antibacterial testing, which must be explored to compare the effect.

## 4. Materials and Methods

### 4.1. Materials

CIP-HCl, sodium borohydride (Sigma Aldrich, St. Louis, MO, USA), silver nitrate (Sigma Aldrich, St. Louis, MO, USA), and nanohybrid resin composite containing Bis-GMA-TEGDMA (Nexcomp) were bought.

### 4.2. Synthesis of Silver Nanoparticles

Silver nanoparticles were synthesized using the chemical reduction method proposed by Mulfinger and coworkers [26]. The reaction involves dropwise addition of 10 mL of 1 mM silver nitrate solution to a 30 mL ice-cold 2 mM sodium borohydride solution, stirring vigorously until the solution turns yellow golden in color. SEM and UV-vis absorption spectroscopy were used to determine the produced particles’ size, shape, and optical characteristics.

### 4.3. Preparation of Ciprofloxacin-Loaded Silver Nanoparticle

CIP-AgNPs were made by adding 0.001M CIP aqueous solution to 100 mL synthesized AgNPs with continuous stirring. To enhance the interaction between CIP and AgNPs, ultrasonication was performed to enhance the interaction of the drug with the NPs [20].

### 4.4. Characterizations of AgNPs & Ciprofloxacin Loaded AgNPs

The confirmation of the synthesis of AgNPs and CIP-AgNPs was determined using a UV-2800 spectrophotometer (Madrid, Spain) to detect the SPR peak. For surface characterization, Bruker FTIR Spectrometer ALPHA ll (Westborough, MA, USA) was used to confirm the loading of CIP on AgNPs. Using SEM, the surface morphology was examined (Tescan, Brno, Czech Republic). Zetasizer (Malvern, VIC, Australia) was used to examine the zeta potential of synthesized AgNPs.

### 4.5. Formulation of Experimental Resin Composites

Nexcomp-META BIOMED containing Bis-GMA, UDMA, Bis-EMA organic polymers (40:40:30), and 70% by weight barium aluminum silicate nanofillers was used to prepare experimental resin composites.

A total of 3 groups of experimental resin composite discs with unmodified composite discs, composite discs containing 1% AgNPs, and composite discs containing 1% CIP-AgNPs were prepared using a plastic mold. These were cylindrical shaped discs that were 4 mm in diameter and 2 mm in height. NPs were incorporated into resin composite at room temperature and were homogenized manually [27]. Composite discs were then cured for 2 h using blue UV light of 400 mV/cm^2^ intensity with a 430–480 nm wavelength.

### 4.6. Isolation of Bacterial Strains

There were 3 antibacterial activity models for *Enterococcus faecalis*, *Streptococcus mutans,* and Saliva microcosm. To isolate bacterial strains, 10 saliva samples were collected from people who had never had orthodontic treatment; those samples were mixed and serially diluted. After serial dilution, the sample was spread on tryptic soy agar (TSA) plates, then incubated. These plates were checked for bacterial growth after 24 h. *Enterococcus faecalis* and *Streptococcus mutans* colonies were differentiated morphologically. For confirmation, these colonies were streaked on blood agar plates and were isolated after incubation. For the microcosm model, an inoculum was prepared by the saliva sample dilution with sterile glycerol of 30% in concentration [28].

### 4.7. Antibacterial Activity

The bacterial growth medium was tryptic soy broth with 1 percent sucrose (TSBS). Overnight, the preculture was incubated. On the other hand, a culture derived from this preculture was incubated for 3 h. The culture was serially diluted once the optical density (OD) reached 1 at 600 nm. To prepare an antibacterial activity model, 500 µL of bacterial suspension was added in Eppendorf along with experimental composite discs that were incubated for 8 h. Then, 50 µL of incubated samples was spread on TSA plates that were incubated for 18 h. After 18 h of incubation, the number of colonies was counted. CFU/mL was calculated using the formula CFU/mL = (no. of colonies × dilution factor)/volume of culture plate. Antibacterial activity of pure composite discs and 1% AgNP-containing composite discs were used as control groups. The experiment was performed in triplicate.

### 4.8. Biocompatibility Analysis of Experimental Resin Composites

The hemolytic assay was performed to check the biocompatibility of experimental dental composite discs containing CIP-AgNPs, AgNPs, and commercially bought resin composite. Human blood samples were taken. As a positive control. Triton X-100 (0.5%) was used as a positive control, while PBS was used as a negative control. The absorbance peak was calculated at 545 nm using UV-vis Spectrophotometer. The percentage hemolysis was calculated using the formula below [23].
%Hemolysis = (Sample OD − Negative Control OD/Positive Control OD − Negative Control OD) × 100(1)

### 4.9. Compressive Strength (CS.)

The composite specimens were kept at 37 °C in the water bath for 24 h. The compressive strength was investigated using Universal Testing Machine UTM at a crosshead speed of 0.5 cm/min and a load cell of 5KN. CS values were calculated using the formula below.
Compressive strength (MPa) = Failure Load (N)/Area (mm^2^)(2)

### 4.10. Statistical Analysis

Graph Pad Prism 9.2 was used for statistical analysis. Wilcoxon T-tests were used to compare groups at a significance level of 95%.

## 5. Conclusions

The results indicated that the antibacterial activity and compressive strength of resin composites containing CIP-AgNPs were superior to the simple resin composites and were more biocompatible than the resin composites containing AgNPs. Furthermore, they did not negatively affect the aesthetics. Hence, CIP-AgNPs are a good option to enhance the antibacterial activity of dental resin composites and improve the longevity of tooth restorations. These findings demonstrated solid foundational applications for dental composite resins modified with CIP-AgNPs. The long-term antibacterial effect of CIP-AgNPs in resin composites should be addressed in future studies. 

## Figures and Tables

**Figure 1 molecules-27-07182-f001:**
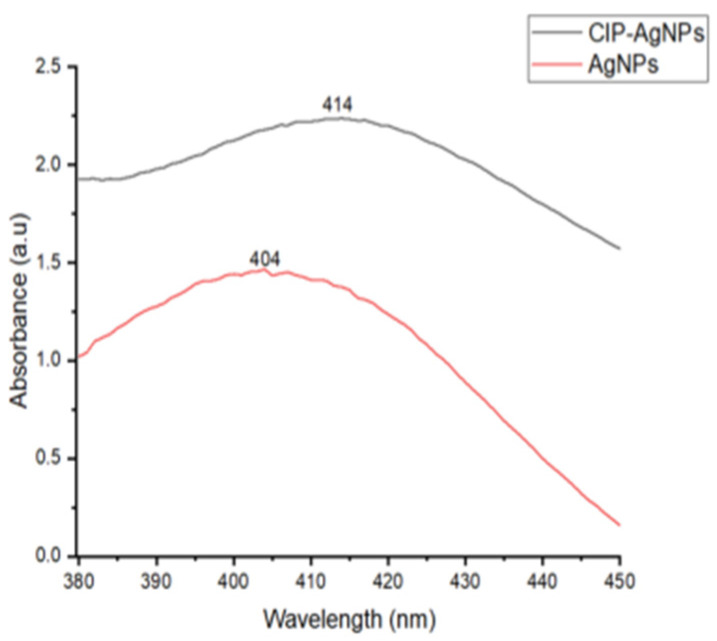
UV- Spectrograph of AgNPs & CIP-AgNPs.

**Figure 2 molecules-27-07182-f002:**
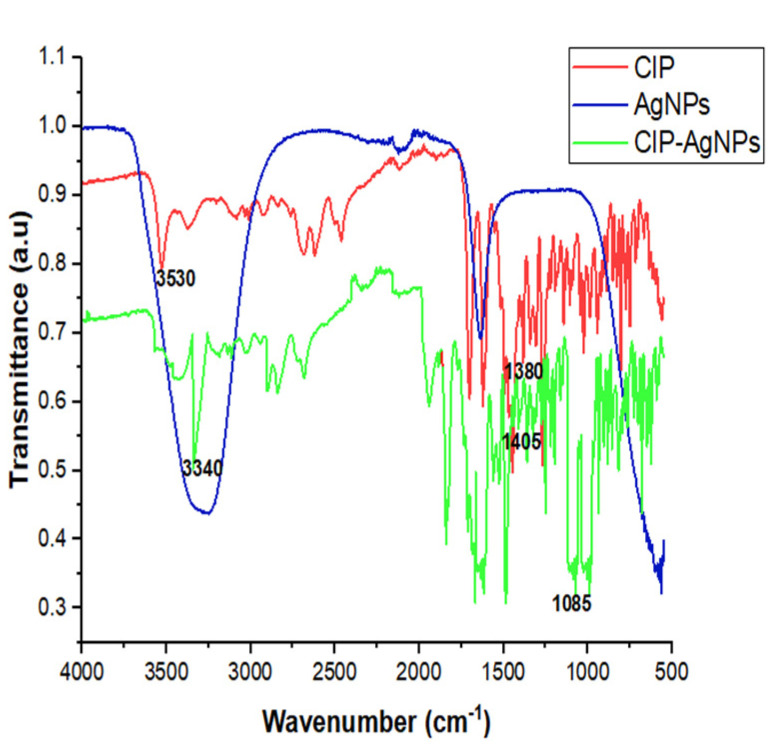
FTIR Spectra of CIP, AgNPs and CIP-AgNPs.

**Figure 3 molecules-27-07182-f003:**
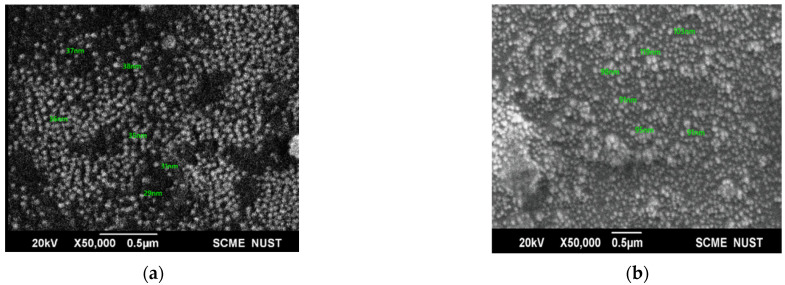
(**a**) SEM image of AgNPs; (**b**) SEM image of CIP-AgNPs.

**Figure 4 molecules-27-07182-f004:**
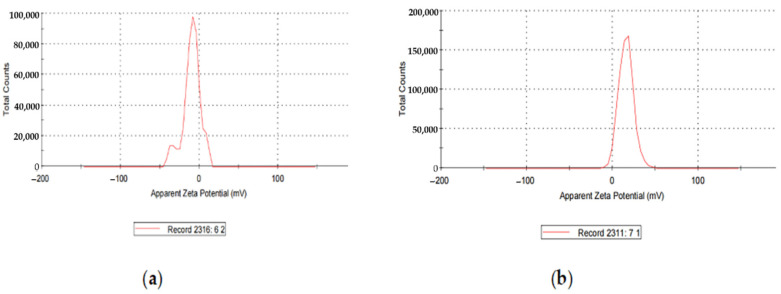
(**a**) Zeta Potential of AgNPs; (**b**) zeta Potential of CIP-AgNPs.

**Figure 5 molecules-27-07182-f005:**
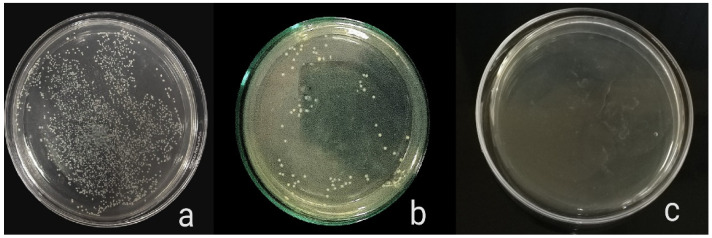
*E. faecalis* growth of unmodified composite discs (**a**), composite discs with 1% AgNPs (**b**), and composite discs with 1% CIP-AgNPs (**c**).

**Figure 6 molecules-27-07182-f006:**
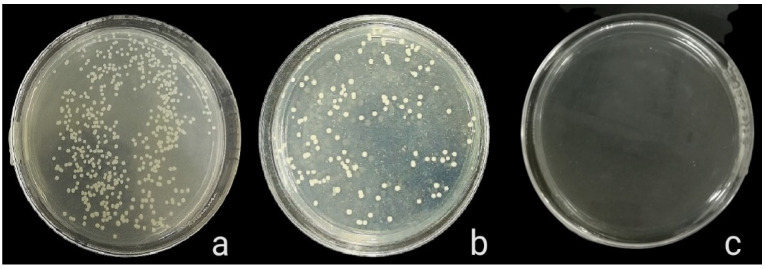
*S. mutans* growth of unmodified composite discs (**a**), composite discs with 1% AgNPs (**b**), and composite discs with 1% CIP-AgNPs (**c**).

**Figure 7 molecules-27-07182-f007:**
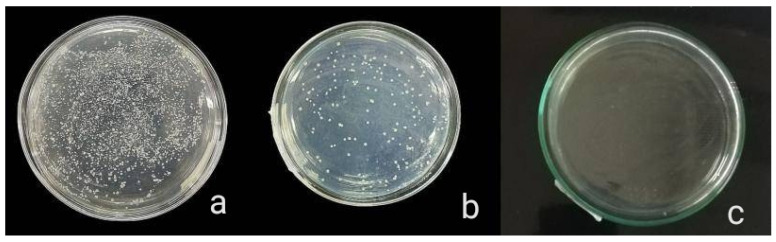
Saliva microcosm bacterial growth of unmodified composite discs (**a**), composite discs with 1% AgNPs (**b**) and composite discs with 1% CIP-AgNPs (**c**).

**Figure 8 molecules-27-07182-f008:**
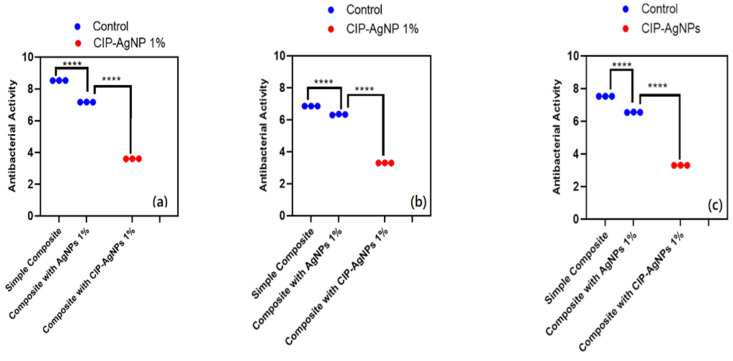
CFU/mL of (**a**) *E. faecalis*, (**b**) *S. mutans*, (**c**) Saliva microcosm model, antibacterial effect of modified composite resin discs (*p* < 0.05). **** *p* < 0.0001.

**Figure 9 molecules-27-07182-f009:**
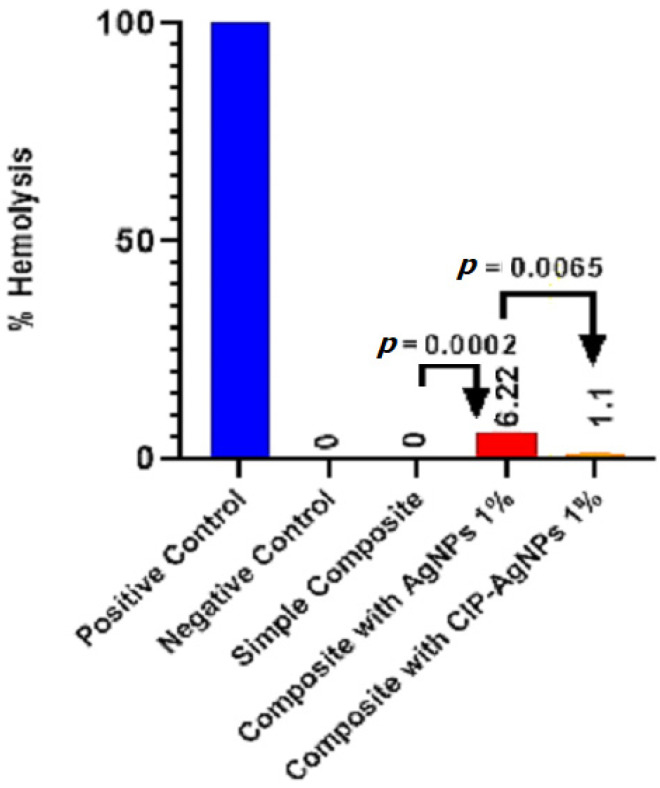
Hemolytic activity of experimental composite discs.

**Figure 10 molecules-27-07182-f010:**
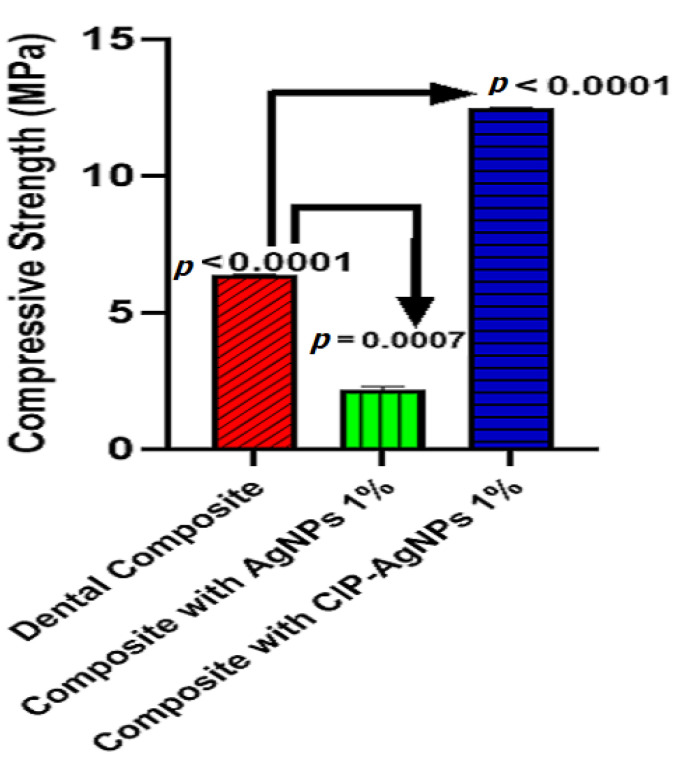
Compressive strength of experimental composite discs.

**Table 1 molecules-27-07182-t001:** FTIR wavenumbers of CIP, AgNPs and CIP-AgNPs.

Functional Groups	Wavenumber (cm^−1^)		
	AgNPs	CIP	CIP-AgNPs
O-H	3345	3530	3340
C-H	-	-	2923
C=O	1636	1635	1636
C=C	666	667	667
C-F	-	1080	1085
N-H	-	1380	1045

## Data Availability

Not applicable.

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
