# Peer review of "Antibacterial Activity of Dental Composite with Ciprofloxacin Loaded Silver Nanoparticles"

_molecules, 2022, doi:10.3390/molecules27217182_

Round 1

Reviewer 1 Report

This is very promising work with strong potential for application of the developed system which can be accepted for publication after major revision. The following issues should be addressed. 

The quality of Fig. 1, 4, and 8 are ultimately low and must be improved. 

Please present the results of the FTIR spectroscopy in form of the Table with wavenumbers of picks and assigned groups.

Discussion about the impact of the functionalization of AgNPs by Ciprofloxacin on absorption maxima at UV spectroscopy should be provided.

The information about the size of the AgNPs and CIP-AgNPs should be provided in the text.

Discussion about the impact of the functionalization of AgNPs by Ciprofloxacin on zeta potential should be provided.

How much Ciprofloxacin was uploaded by AgNPs? What is the mechanism of the interaction between Ciprofloxacin and AgNPs? Appropriate discussion should be provided.

What is the chemical structure of the Nexcomp-META BIOMED? What about the distribution of the CIP-AgNPs in the polymer matrix?

Please add a, b and c to Fig. 5-7 and their captions.

What was the antibacterial mechanism, if I am right, silver ions and Ciprofloxacin were released and gave a synergetic effect?  Appropriate discussion should be provided.

Please cite relevant papers where similar systems were described:

https://doi.org/10.1016/j.colsurfa.2022.128525

https://doi.org/10.1016/j.fbio.2022.101867

https://doi.org/10.1016/j.msec.2019.109806

Author Response

Response to Reviewer 1:

It is very promising work with strong potential for application of the developed system which can be accepted for publication after major revision. The following issues should be addressed.

Response to comments: Thank you very much for taking out the time to review this manuscript. All queries have been addressed point-by-point, the manuscript is now much improved thanks to valuable comments and suggestions. All the changes are traceable within the manuscript

The quality of Fig. 1, 4, and 8 are ultimately low and must be improved.

Response to comments: Figures 1, 4 and 8 are updated as suggested.

Please present the results of the FTIR spectroscopy in form of the Table with wavenumbers of picks and assigned groups.

Response to comments: Table has been added as suggested (line 78).

Discussion about the impact of the functionalization of AgNPs by Ciprofloxacin on absorption maxima at UV spectroscopy should be provided.

Response to comments: The absorption maxima has been mentioned in line 72.

The information about the size of the AgNPs and CIP-AgNPs should be provided in the text.

Response to comments: The information has been added in line 86 as suggested.

Discussion about the impact of the functionalization of AgNPs by Ciprofloxacin on zeta potential should be provided.

Response to comments: The information has been added in line 96 as suggested. It gave positive charge to the surface of the NPs.

How much Ciprofloxacin was uploaded by AgNPs? What is the mechanism of the interaction between Ciprofloxacin and AgNPs? Appropriate discussion should be provided.

Response to comments: Loading capacity has not been measured and added as limitation in  (line 84). Mechanism of Interaction of CIP with AgNPs are added in line 150.

What is the chemical structure of the Nexcomp-META BIOMED? What about the distribution of the CIP-AgNPs in the polymer matrix?

Nexcomp-META BIOMED  is a commercial composite with three different polymers and Composition has been added.   (line 214). Distribution of NPs in composites has been added as limitation (line 84)

Please add a, b and c to Fig. 5-7 and their captions.

Response to reviewer comment: Figures are updated as suggested.

What was the antibacterial mechanism, if I am right, silver ions and Ciprofloxacin were released and gave a synergetic effect?  Appropriate discussion should be provided.

YES, you are right. There is an additional or synergistic effect of the Cipro antibiotic drug and Ag+ (silver NPs having inherent antibacterial effects/activity). Discussion has been updated as suggested

Please cite relevant papers where similar systems were described:

https://doi.org/10.1016/j.colsurfa.2022.128525

https://doi.org/10.1016/j.fbio.2022.101867

https://doi.org/10.1016/j.msec.2019.109806

Reviewer 2 Report

In the manuscript entitled “Antibacterial Activity of Dental Composite Discs integrated with Ciprofloxacin loaded Silver Nanoparticles” the preparation and properties of the dental material are described. The manuscript is well prepared and organized, the procedures are described in a way allowing repetition in another laboratory. The prepared material is consequently analyzed leading to conclusions supported by experimental data.

Thus, I recommend publication the manuscript after major revision, as indicated:

11.       size of CIP-AgNP and Ag nanoparticles should be determined on the basis of SEM images or other methods. Exact determination of size is missing. The difference between Ag NP and CIP-Ag NP is barely visible. There is also lack of information on aggregation of particles. The low negative or positive charge may suggest the poor distribution of individual particles (also in resin matrix).

22.       Fig.4 and Fig.8 barely visible.

33.       Fig.5,6,7 – the images should be numbered a,b,c.

44.       Discussion is superfluous since it is a repetition of the results of the research presented in an earlier chapter.

Author Response

In the manuscript entitled “Antibacterial Activity of Dental Composite Discs integrated with Ciprofloxacin loaded Silver Nanoparticles” the preparation and properties of the dental material are described. The manuscript is well prepared and organized, the procedures are described in a way allowing repetition in another laboratory. The prepared material is consequently analyzed leading to conclusions supported by experimental data.

Thus, I recommend publication the manuscript after major revision, as indicated:

Response to comments: Thank you very much for taking out the time to review this manuscript. All queries have been addressed point-by-point, the manuscript is now much improved thanks to valuable comments and suggestions. All the changes are traceable within the manuscript

 size of CIP-AgNP and Ag nanoparticles should be determined on the basis of SEM images or other methods. Exact determination of size is missing. The difference between Ag NP and CIP-Ag NP is barely visible. There is also lack of information on aggregation of particles. The low negative or positive charge may suggest the poor distribution of individual particles (also in resin matrix).

We used image j software for particle size analysis.  Our aim was to conduct the morphology analysis using STEM, which is better method for size evaluation of small particles however, the facility is not available so we did SEM. Information regarding aggregation has been added

  Fig.4 and Fig.8 barely visible.

Response to reviewer comment: Figures are updated as suggested.

Fig.5,6,7 – the images should be numbered a,b,c.

Response to reviewer comment: Figures are updated as suggested.

  Discussion is superfluous since it is a repetition of the results of the research presented in an earlier chapter.

Response to reviewer comment: Discussion has been updated as suggested. Information regarding antibacterial activity and mechanism of action has been added. Once again thanks for your fruitful comments. We tried to cover all the suggestions mentioned. Thank you.

Reviewer 3 Report

The Authors present the results of their research on the antibacterial activity of dental composites with ciprofloxacin loaded silver nanoparticles (AgNPs). These idea could be interesting, unfortunatelly in a present form the manuscript  looks more like a student's report than the research article.

1) The introduction is very scam, it is based on 9 references which surely do not exhaust the broad research area of dental composites and antibacterial activity of AgNPs

2) The experimental research are shown but no discussion and conclusions are provided

3) The quality of SEM images (Fig. 3) is poor, to elaborate on size of nanoparticles and their coating with ciprofloxacin the images should be recorded with higher magnification.  What is the size of NPs? Is the two-layer composition of coated NPs visible? What is the thickness of the ciprofloxacin layer?

4) Presented results show that all properties of composites, including the mechanical strength, are better for materials with ciprofloxacin coated AgNPs comapred to pure AgNPs. To verify whether this effect results from the coating of AgNPs or just the ciprofloxacin itself, the control experiments should be performed, with dental composites with only ciprofloxacin added (no AgNPs).

Author Response

The Authors present the results of their research on the antibacterial activity of dental composites with ciprofloxacin loaded silver nanoparticles (AgNPs). These idea could be interesting, unfortunatelly in a present form the manuscript  looks more like a student's report than the research article.

Response to comments: Thank you very much for taking out the time to review this manuscript. Your concern is true and while keeping this comment in consideration we have updated the manuscript. I hope the current form of paper is in better shape.

1) The introduction is very scam, it is based on 9 references which surely do not exhaust the broad research area of dental composites and antibacterial activity of AgNPs

Response to comments: Introduction has been updated considering this comment. More details are added as suggested.

2) The experimental research are shown but no discussion and conclusions are provided

Response to comments: Conclusion is provided in the line 261 and discussion has been updated.

3) The quality of SEM images (Fig. 3) is poor, to elaborate on size of nanoparticles and their coating with ciprofloxacin the images should be recorded with higher magnification.  What is the size of NPs? Is the two-layer composition of coated NPs visible? What is the thickness of the ciprofloxacin layer?

Response: The average sizes are measured using image j software and average is now reported (line 86). The thickness cant be measured due to low resolution, however, the increase in size confirmed the coating. Also other characterization techniques confirmed the loading.

4) Presented results show that all properties of composites, including the mechanical strength, are better for materials with ciprofloxacin coated AgNPs comapred to pure AgNPs. To verify whether this effect results from the coating of AgNPs or just the ciprofloxacin itself, the control experiments should be performed, with dental composites with only ciprofloxacin added (no AgNPs).

Response: AgNPs exhibits inherent antibacterial activity which is enhanced after loading with antibiotics as presented in literature.

2- Zafar N, Uzair B, Menaa F, Khan BA, Niazi MBK, Alaryani FS, Majrashi KA, Sajjad S. Moringa concanensis-Mediated Synthesis and Characterizations of Ciprofloxacin Encapsulated into Ag/TiO2/Fe2O3/CS Nanocomposite: A Therapeutic Solution against Multidrug Resistant E. coli Strains of Livestock Infectious Diseases. Pharmaceutics. 2022 Aug 17;14(8):1719. doi: 10.3390/pharmaceutics14081719. PMID: 36015345; PMCID: PMC9412270.

3-Zafar N, Uzair B, Niazi MBK, Menaa F, Samin G, Khan BA, Iqbal H, Menaa B. Green Synthesis of Ciprofloxacin-Loaded Cerium Oxide/Chitosan Nanocarrier and its Activity Against MRSA-Induced Mastitis. J Pharm Sci. 2021 Oct;110(10):3471-3483. doi: 10.1016/j.xphs.2021.06.017. Epub 2021 Jun 11. PMID: 34126118.

However, your point is also valid. The point that has been mentioned is a limitation from our side and has been mentioned in limitation section.Once again thanks a lot for the fruitful comments.

Round 2

Reviewer 1 Report

The authors have answered most of my concerns but the quality of the Figs. 1 and 4 is still low and that should be improved.   

Author Response

Kindly check the attached file. 

Reviewer 2 Report

The manuscript can be published in a present form.

Author Response

Kindly check the attached file. 

Reviewer 3 Report

No additional comments.

Author Response

Kindly check the attached file. 
